# Novel Concept of Alpha Satellite Cascading Higher-Order Repeats (HORs) and Precise Identification of 15mer and 20mer Cascading HORs in Complete T2T-CHM13 Assembly of Human Chromosome 15

**DOI:** 10.3390/ijms25084395

**Published:** 2024-04-16

**Authors:** Matko Glunčić, Ines Vlahović, Marija Rosandić, Vladimir Paar

**Affiliations:** 1Faculty of Science, University of Zagreb, 10000 Zagreb, Croatia; vpaar@hazu.hr; 2Algebra LAB, Algebra University College, 10000 Zagreb, Croatia; ines.vlahovic@algebra.hr; 3Department of Internal Medicine, University Hospital Centre Zagreb, 10000 Zagreb, Croatia; rosandic@hazu.hr; 4Croatian Academy of Sciences and Arts, 10000 Zagreb, Croatia

**Keywords:** T2T-CHM13 human assembly, alpha satellites, higher-order repeats, centromere, GRM algorithm

## Abstract

Unraveling the intricate centromere structure of human chromosomes holds profound implications, illuminating fundamental genetic mechanisms and potentially advancing our comprehension of genetic disorders and therapeutic interventions. This study rigorously identified and structurally analyzed alpha satellite higher-order repeats (HORs) within the centromere of human chromosome 15 in the complete T2T-CHM13 assembly using the high-precision GRM2023 algorithm. The most extensive alpha satellite HOR array in chromosome 15 reveals a novel cascading HOR, housing 429 15mer HOR copies, containing 4-, 7- and 11-monomer subfragments. Within each row of cascading HORs, all alpha satellite monomers are of distinct types, as in regular Willard’s HORs. However, different HOR copies within the same cascading 15mer HOR contain more than one monomer of the same type. Each canonical 15mer HOR copy comprises 15 monomers belonging to only 9 different monomer types. Notably, 65% of the 429 15mer cascading HOR copies exhibit canonical structures, while 35% display variant configurations. Identified as the second most extensive alpha satellite HOR, another novel cascading HOR within human chromosome 15 encompasses 164 20mer HOR copies, each featuring two subfragments. Moreover, a distinct pattern emerges as interspersed 25mer/26mer structures differing from regular Willard’s HORs and giving rise to a 34-monomer subfragment. Only a minor 18mer HOR array of 12 HOR copies is of the regular Willard’s type. These revelations highlight the complexity within the chromosome 15 centromeric region, accentuating deviations from anticipated highly regular patterns and hinting at profound information encoding and functional potential within the human centromere.

## 1. Introduction

Until recently, the centromeric region of the human genome largely remained uncharted, earning it the moniker of the genome’s ‘black hole’. Recent technological advancements have catalyzed comprehensive investigations into the structure and function of the centromeric region within the complete human genome assembly, known as T2T-CHM13 [1,2,3,4,5,6]. This has prompted intense studies of higher-order repeats in complete genomic assemblies [4,7,8,9].

In the last century, with limited sequencing available, researchers discovered that human centromeres consist mostly of approximately 171 bp alpha satellite repeat monomers. These monomers are organized into sequences of *n* monomers, collectively known as *n*mer HORs [10,11,12,13,14]. The degree of divergence between any two monomers within each HOR copy is significant, ranging from approximately 20% to 40%. However, HOR copies are further organized in tandem, with minimal divergence between HOR copies, typically less than 5%. Monomers exhibiting less than 5% divergence are classified as the same monomer type. Willard and colleagues found that, within each HOR copy, the monomers belong to different monomer types. This distinct pattern is known as ‘Willard’s type HORs’ and was the subject of extensive study based on the limited sequencing data previously available in the centromeric region [4,8,9,15,16,17,18,19,20,21,22,23,24,25,26,27,28,29,30,31,32,33,34,35]. In a Willard’s type *n*mer HOR array, the most common HOR copy with *n* constituting monomers is referred to as canonical. Copies in the same HOR array that contain inserts or deletions compared to the canonical HOR copy are known as variants.

The availability of large complete assemblies featuring long higher-order repeat (HOR) units necessitates the development of specific tools for precise identification and annotation. While various existing software applications effectively identify regions with tandem repeats (e.g., [36,37,38]), they fall short of providing precise annotations for individual repeat locations or HORs. Similarly, more recent tools designed for annotating human alpha satellite HORs within genomic sequences (e.g., [39,40,41,42,43]) have limited broader applicability [8]. On the other hand, a specific set of software has been developed for the accurate identification of Willard’s type HORs [8,18,28,44]. In the context of the complete assembly of human chromosome 15, alpha satellite HORs were initially computed using the NTRprism algorithm [4], which bears resemblance to the 2007 version of GRM [44].

In this study, we introduce a novel algorithm named GRM2023, an enhanced iteration of our earlier algorithm, the Global Repeat Map (GRM) [18,44]. GRM2023 is specifically designed to characterize HORs extending beyond the scope of Willard’s classification, denoted as cascading higher-order repeats (cascading HORs), wherein certain monomer types are repeated within a canonical HOR copy. It is important to acknowledge that providing a comprehensive description of the structural organization of alpha satellite higher-order repeats poses a significant challenge, and discrepancies exist among various methodologies (for further details, refer to subsequent sections, including Materials and Methods and Conclusion). However, both GRM and GRM2023 tools offer the advantage of high precision in identifying HOR copies within HOR arrays. Additionally, GRM2023 not only identifies peaks corresponding to repeated DNA sequences but also discerns peaks associated with repeats that are not tandemly arranged. Subsequently, in the second step, GRM2023 verifies whether these identified repeats indeed represent tandem repeats.

Southern analysis of human chromosome 15 unveiled an alpha satellite higher-order genomic band of 2.5 kb, roughly equivalent to a 15mer, in the case of pTRA-20. The analysis also detected multiple higher-order bands at 3.5 kb (~20mer), 4.5 kb (~26mer), and 5 kb (~29mer) [45]. Additional Southern blot analyses further corroborated the identification of alpha satellite subfamily HORs with both 2.5 kb (~15mer) and 1.8 kb (~11mer) components [21]. It was inferred that the pTRA-20 clone represented a subfragment of the 2.5 kb HOR, with an additional 0.7 kb sequence (~4mer) adjacent to the 1.8 kb pTRA-20 sequence. These data emphasize the intricate organization of the pTRA-20 specific alpha satellite DNA, suggesting the presence of differentially sized HORs [45].

Two recent analyses of the T2T-CHM13 human genome assembly have recently been reported for human chromosome 15. The alpha satellite 11mer, 20mer, and 26mer HORs were identified using the NTRprism algorithm [4]. It was observed in Ref. [4] that the NTRprism algorithm is similar to the method previously employed [44] for the detection of Willard’s type HORs with only one monomer of each type in canonical HOR copies.

In the computation using the HICAT algorithm, three HORs were reported: R1L11—11mer, R2L4—4mer and R3L22—22mer [42]. It was also noted that the R1L11 unit contained also HOR copy variants with a monomer composition of 1–2–3–4–5–(6–7–8–9)×n–10–11 (each number represents a monomer, and “×n” represents the number of times that a defined monomer set was repeated). For example, four consecutive monomers, 6–7–8–9, in the R1L11 unit repeated, and most of monomer repeats also existed according to Figure 4e in [42]. The HOR copy sample 1–2–3–4–5–(6–7–8–9) × n–10–11 for n = 2 is then 15mer HOR with an internal repeat of the monomer group (6–7–8–9). This means that HICAT predicts that these HOR copies are 15mer variants of 11mer HOR and not canonical 15mer HOR.

Here, using our novel precise GRM2023 algorithm, we unveil the identification and characterization of complex alpha satellite HORs in the T2T-CHM13 genome assembly of human chromosome 15. We identified a complex GRM spectrum in the T2T-CHM13 human genome assembly precisely revealing the cascading 15mer and 20mer, Willard’s type 18mer, and interspersed Willard’s type 25mer/26mer HORs. We also identified subfragment monomeric repeats from *n*mer HORs, with a periodicity of 4, 7, and 11 for 15mer HOR, 5 and 15 for 20mer HOR, and 34 for 25mer HOR.

## 2. Results and Discussion

### 2.1. GRM (Global Repeat Map) Diagram and MD (Monomer Distance) Diagram for T2T-CHM13 Assembly of Human Chromosome 15

In the initial step, we identify tandemly organized alpha satellite monomers within the T2T-CHM13 genomic assembly of human chromosome 15. This process yielded an assembly of 15,458 tandemly arranged monomers, enumerated sequentially from 1 to 15,458 according to their order of occurrence. Subsequently, employing the high-precision GRM2023 algorithm, we compute the corresponding GRM (Global Repeat Map) diagram for this array of tandemly organized monomers. The GRM2023 algorithm represents a novel and enhanced version (refer to the Methods section) of the Global Repeat Map (GRM) algorithm previously utilized for identifying Willard’s type HORs, which lack duplication of any monomer type within an HOR copy [18,28,46,47]. A peak of period *n* corresponds to *n* × 171 bp (previously referred to as fragment length [18,28]. GRM2023 precisely identifies HORs beyond the Willard’s type HORs, identifying HORs with more than one monomer of the same type within an HOR copy. Such HORs will be referred to as cascading higher-order repeats (cascading HORs). The GRM2023 algorithm also identifies the intra- and inter- HOR-copy monomer repeats, referred to as HOR subfragments (SF). The GRM diagram computed for the T2T-CHM13 assembly of human chromosome 15 reveals alpha satellite GRM peaks at periods 4, 5, 7, 11, 15, 18, 20, 25, 26 and 34 units (units of ~171 bp) (Figure 1a). These GRM peaks correspond to repeats in the genomic subspace of tandemly repeated alpha satellite monomers. However, not all these GRM peaks correspond to HOR copies, as observed from the MD diagram (Figure 1b).

The MD diagram displays the relationship between period and monomer enumeration (Figure 1b). Each monomer enumeration (horizontal axis) and the corresponding distance to the next monomer of the same type in tandemly organized monomer sequence (vertical axis) determine coordinates of a point in MD diagram, referred to as MD points. All MD points corresponding to an HOR are situated on a horizontal MD line segment, which possesses a vertical coordinate equal to the HOR’s period. These MD points cluster so densely along the line segment that they form what appears to be a continuous line across the interval comprising the constituent monomers. For beyond the Willard’s type of HORs, additional MD line segments appear beneath the primary HOR’s MD line segment within the same monomer enumeration interval, featuring periods shorter than that of the original HOR copies. Such interspersed repeats within the HOR array, both within and between HOR copies, are termed subfragments. They are defined by the same type of monomer at both ends of subfragment. For example, the GRM peak of period 15 and its MD line segment correspond to the 15mer HOR array while the GRM peaks at periods 4, 7 and 11 and the corresponding MD line segments correspond to its subfragments, as will be shown later. Conversely, the MD line segments at periods 15/16 and 18 correspond to the Willard’s type 15mer/16mer and 18mer HORs, respectively, without subfragments. The locations of identified HOR arrays in human chromosome 15 are illustrated in the ideogram in Figure 2.

### 2.2. Aligned Scheme for Cascading 15mer HOR Array with 4mer, 7mer and 11mer Subfragments

The GRM peak at period 15 (in 171 bp units) in Figure 1a signifies the 15mer HOR, forming an HOR array designated as hor3. Correspondingly, in the MD diagram shown in Figure 1b, the related MD line segments span the interval of tandemly repeated monomers between enumeration ~9300 and ~15,200 at period 15 on the vertical axis, thereby delineating a 15mer HOR. Within this same monomer enumeration interval, there exist equidistant MD line segments for periods 4, 7 and 11, which represent subfragments of the 15mer HOR. These subfragments are indicative of both inter-monomer and intra-monomer segments within the 15mer HOR array. The MD line for the period 15, spanning from around enumeration ~6000 to ~9300, does not stem from a 15mer HOR. Instead, it, along with the MD line for the period 5, constitutes sub-fragments of the 20mer HOR.

The canonical alpha satellite 15mer HOR copy comprises 15 monomers of 9 distinct types, arranged in a cascading scheme across four rows. This arrangement aligns the 15 monomers in cascading columns by their types (Figure 3a), as opposed to a linear, single-row presentation that follows the sequence in which the monomers appear (Figure 3b).

In the process of constructing HOR copy schemes, as one progresses through the sequence of monomers within the HOR array, if the next monomer encountered is of the same type as one previously placed in the current row, this monomer is then positioned in the row below and aligned to the corresponding column of the monomer type.

Such an arrangement for canonical HOR copy involves the repetition of some monomer types, with monomers of the same type positioned in different rows, as illustrated for canonical 15mer HOR scheme in Figure 3a: three monomers of the type t2 in the second column, three monomers of the type t3 in the third column, two monomers of the type t1 in the first column and two monomers of the type t4 in the fourth column. In this way, each of the first and second rows in canonical 15mer HOR copy contains one monomer of the type t1, while the first, third and fourth rows each contain a monomer of type t3. However, within each row, there are no repetitions of any monomer type.

In the hor3 array, 279 of 429 15mer HOR copies (65%) are canonical, while the remaining 150 HOR copies (35%) are variants. Among the variant HOR copies, pronounced variants include those characterized by (1) omitting the entire first row of the canonical HOR copy or (2) partially reshuffling other rows. Conversely, the shortest variant consists of four monomers of types t1–t4, identical to the quadruplet found in the first row of the canonical HOR copy. The complete HOR alignment scheme for the cascading 15mer alpha satellite HOR hor3, computed using the GRM2023 algorithm on the T2T-CHM13 complete genomic assembly of human chromosome 15, is presented in Appendix A. The corresponding consensus sequences are provided in Appendix A.

Let us briefly discuss the manner in which these complex patterns (Appendix A) might be generated. When examining the spatial distribution of these 15mer HORs, we notice that the series of HORs begins with variants b and b’, followed by variants a’ and c. At the beginning of the series, the canonical HOR (a) only occasionally appears. As we move towards the middle of the HOR series, the canonical HOR overwhelmingly dominates, with significant interjections of variants a’, c, and c’ around the middle, occasionally accompanied by variants d and d’. Variant e of the HOR (first four monomers, m1–m4) continuously appears as an occasional byproduct of duplicating the canonical copy. After the middle of the HOR series, the canonical HOR copy strongly predominates again, with slight interjections of variants a’ towards the end. Hypothetically, we can conclude that the pure, canonical HOR copy duplicated from two centers located around the positions of the first and third quarters of the HOR series. In the area around these two centers, duplication is nearly perfect. However, as we move further away from the duplication centers, the copies become increasingly imperfect, leading to the emergence of more variant HORs. An interesting fact is that the deviation from the perfect canonical HOR always occurs in a limited set of represented variant HORs (Figure 3c), unlike the expected stochastic errors resulting from duplications, insertions, or deletions of any individual monomers or entire groups of monomers.

In the case of the Willard’s type nmer HOR array, the corresponding prominent GRM peak emerges at a period equal to the length of the HOR copy, i.e., ~*n* × 171 bp, because the divergence between HOR copies is much smaller than the divergence between constituting monomers of the same HOR copy. On the other hand, in the case of the cascading HOR array, additional substantial GRM peaks, obtained through the use of GRM2023 algorithm, also appear also due to repeats of additional monomers of the same type within HOR copies. This is because they exhibit small divergence with other monomers of the same type, giving rise to additional GRM peaks that are absent in the case of Willard’s HOR arrays. The GRM2023 algorithm also identifies additional intra-HOR-copy and inter-HOR-copy monomer repeats between two neighboring HOR copies (Figure 4) of cascading HORs. In the caption to Figure 4, the corresponding intra-HOR-copy monomer distances of the same type are depicted as $d(1_{1}$, $1_{2})$ = 4, $d(2_{1}$, $2_{2})$ = 4, $d(3_{1}$, $3_{2})$ = 7, $d(4_{1}$, $4_{2})$ = 11, $d\left(3_{2},3_{3}\right)=$ 4, $d(2_{1}$, $2_{3})$ = 11, $d(2_{2}$, $2_{3})$ = 7, and the inter-HOR-copy monomer distances as ${d(1}_{2}$, $1_{1}^{,})$ = 11, ${d(2}_{2}$, $2_{1}^{,})$ = 11, ${d(2}_{3}$, $2_{1}^{,})$ = 4, ${d(3}_{3}$, $3_{1}^{,})$ = 4, ${d(4}_{2}$, $4_{1}^{,})=$ 4, ${d(3}_{3}$, $3_{2}^{,})=$ 11. These repeats generate distances of 4, 7, and 11 monomer units, resulting in the MD lines of periods 4, 7 and 11, respectively, within the hor3 array region.

### 2.3. Aligned Scheme for Cascading 20mer HOR Array with 5mer and 15mer Subfragments

In the monomer enumeration region h4, spanning from ~6000 to ~9200 on the MD diagram, three horizontal line segments emerge at periods 20, 15 and 5, corresponding, respectively, to the cascading 20mer HOR (designated as hor4), 15-monomer subfragment and 5-monomer subfragment (Figure 5). The mechanism underlying the formation of these MD lines of periods 5, 15, and 20 within the hor4 array region is elucidated in Figure 6. The canonical 20mer HOR copy comprises 20 monomers of 19 distinct monomer types. This configuration involves the repetition of only one monomer type, t9, as illustrated in Figure 5a (cascading HOR scheme) and Figure 5b (linear HOR scheme). Among all 20mer HOR copies within hor4, 94% conform to the canonical type. In the canonical HOR copy, monomers are arranged in a cascading scheme across two rows, such that the only repeated monomers of type t9 in the first and second row are aligned (Figure 5a).

The alignment scheme for the entire cascading 20mer HOR (hor4) is illustrated in Appendix A, and the corresponding consensus sequences are presented in Appendix A. Some variant HORs of 20mer HOR are shown in Figure 5c.

### 2.4. Aligned Scheme for Willard’s Type 18mer HOR Array

From the MD diagram (Figure 1b), it is evident that the 18mer HOR is situated within a confined monomer region at monomer enumeration ~3000, constituting an HOR array comprising 12 HOR copies (referred to as hor1). The aligned HOR arrangement of hor1 is depicted in Figure 7. This HOR belongs to the Willard’s type, featuring 18 monomers, each of a distinct type, encompassing 10 canonical 18mer HOR copies along with two variants (14mer and 9mer). The alignment scheme for the entire cascading 18mer HOR (hor4) is illustrated in Appendix A, and the corresponding consensus sequences are presented in Appendix A.

### 2.5. Aligned Scheme for Interspersed Willard’s Type 25/26mer HOR Array and 34-Monomer Tertiary Subfragment

In the region of monomer enumeration from ~3100 to ~4600 (designated as h2), there exists an HOR array denoted as hor2, consisting of 25mer and 26mer HORs along with their interspersed contact domain, as illustrated in Figure 8.

The upper segment of the hor2 array consists of a combination of fourteen 26mer HOR copies and three 25mer HOR copies (see Appendix A). Their arrangement is depicted in Figure 8a, showing four 26mer HOR copies followed by one 25mer HOR copy, then three 26mer HOR copies, two 25mer HOR copies, and finally seven 26mer HOR copies. Within the fourteen 26mer HOR copies, the monomers align with each other, and the same alignment is observed among the monomers in the three 25mer HOR copies. However, when monomer types are sequentially denoted in order of appearance as t1, t2, t3..., the monomers do not align between the 26mer and 25mer HOR copies. Only through adjustment of the monomer enumeration, as illustrated in Figure 8b, can we establish that 22 monomers in the 26mer and 25mer arrays are mutually aligned, while the remaining 7 are not. The 26mer and 25mer HOR copies are presented in Appendix A, and the corresponding consensus sequences are presented in Appendix A.

The lower segment of the hor2 array consists of 48 25mer HOR copies, of which 41 are canonical. Appendix A provides an alignment for these 48 25mer HOR copies of Willard’s type. Within the range spanning from the 6th to the 17th 25mer HOR copies, there are three equidistant doublets, each composed of HOR copies 8/9, 11/12, and 14/15. In the first member of each doublet, monomers m13–m25 (t14–t15, t18–t27, t29) are absent, while in the second member, monomers m1–m3 (t1–t3) are absent. Consequently, the first HOR copy of each doublet contains 13 monomers, while the second HOR copy contains 21 monomers, resulting in an additional GRM peak at 13 + 21 = 34 as a hybrid between two variants (12mer and 22mer) of the 25mer HOR, serving as a tertiary HOR.

## 3. Materials and Methods

The alpha satellite HORs were identified in the human chromosome 15 T2T-CHM13 genomic assembly using the GRM2023 algorithm [18,47,48]. The GRM2023 algorithm is an efficient and robust method specifically designed to detect and analyze very large repeat units, such as HORs, within genomic sequences. The GRM method generates a global repeat map in a GRM diagram, identifying all prominent repeats in a given sequence without any prior knowledge of the repeats. Furthermore, once the consensus repeat unit is determined using GRM2023, it can be further combined with a search for dispersed HOR copies or individual constituting monomers.

Specifically, alpha satellite HORs in this study were identified through the following steps:Using GRMapp version 1.0 (the GRM graphical user interface application is freely available at http://genom.hazu.hr/tools.html, URL (accessed on 14 April 2024)), alpha satellite monomers were identified within the entire human chromosome T2T-CHM13 assembly. GRMapp provides all tandem repeats (TRs) in the analyzed assembly as its output. From the list of all TRs, those with lengths of ~171 bp were selected and subjected to GRM diagram analysis within GRMapp. To be classified as alpha satellite monomers, the GRM diagram must exhibit peaks at ~171 bp and multiples at ~342 bp and ~513 kb, and so on.The extracted alpha satellite monomers were compared to each other, and a divergence matrix was created. From the divergence matrix, monomer families were identified, encompassing all monomers that differ from each other by less than 5%.For each monomer family, a consensus sequence was generated using the stand-alone tool for multiple-sequence alignment, pyabPOA (pyabpoa 1.0.0a0), available at https://github.com/yangao07/abpoa, URL (accessed on 14 April 2024). The consensus sequences for all alpha satellite monomer families are provided in Appendix A.Chromosome 15 T2T-CHM13 assembly was searched with all consensus sequences using the Edlib open-source C/C++ library for exact pairwise sequence alignment [49]. The search was conducted base by base for the entire chromosome, considering both the direct and reverse complement consensus sequences.The results of the search in step (iv) are presented graphically (Figure 1, Figure 2, Figure 3, Figure 4, Figure 5, Figure 6 and Figure 7) such that all monomers of the same family are located in the same column and colored with the same color. As a guideline for constructing rows in a cascading HOR scheme for canonical *n*mer HOR copies with monomer types (t1, t2, t3, …, tτ): we commence with a monomer of type t1 and continue until reaching the monomer of type tτ; alternatively, if a monomer of type tk is followed by a subsequent monomer tm where m<k, the subsequent monomer is aligned in the next row. For a representative example, refer to Figure 3.

## 4. Conclusions

The complexity of the chromosome 15 centromere is underscored by the revelation of unannotated HOR arrays and the internal structure thereof. By employing the novel algorithm GRM2023 on the complete T2T-CHM13 assembly of human chromosome 15, we introduce the innovative concept of cascading alpha satellite HORs. Four distinct HOR arrays have been identified: the newly discovered cascading 15mer HOR, comprising 429 copies; the novel cascading 20mer HOR, with 164 copies; the traditional Willard’s type 18mer HOR, encompassing 12 HOR copies; and the interspersed Willard’s type 25mer/26mer HORs, with a distribution of 51/14 copies (Table 1). These results highlight two significant findings: (1) HOR units containing multiple copies of the same monomer type and (2) interspersed HOR copies within the centromere structure, such as cascading HORs, where adjacent unit repeats originate from the same monomer sequence. Appendix A offer a detailed visual depiction of all HOR copies identified using the GRM2023 algorithm.

The GRM2023 algorithm automatically identifies HORs that contain more than one monomer of the same type as cascading HORs in the monomer distance (MD) diagram through parallel line segments. On the horizontal axis, alpha satellite monomers numbered sequentially (1, 2, 3, …) are arranged according to their appearance in tandem repeats. The vertical axis represents the repeat period, defined as the distance to the closest subsequent monomer of the same type. For instance, consider monomer number 6000. The nearest monomer of the same type following number 6000 is number 6020, which is 20 monomers away; thus, a period of 20 for the appearance of the same monomer type is attributed to monomer number 6000. Consequently, the point with a horizontal coordinate of 6000 and a vertical coordinate of 20 corresponds to monomer 6000. The precise identification of HORs as described herein highlights a significant deviation from tandem regularity in the centromeric sequence, thereby suggesting greater potential for information storage within the centromere.

It should be noted that the HOR annotation method employed by the NTRprism algorithm, as outlined in [4], closely resembles the 2007 version of GRM [44], which was specifically tailored for the identification of Willard-type HORs. Consequently, this method proves less effective for identifying more complex HOR structures, such as cascading HORs. Notably, our application of the enhanced GRM2023 algorithm has uncovered the largest HOR within hor3 to be a 15mer (illustrated in Figure 1b and Appendix A), a significant revision from its previous classification as an 11mer by NTRprism [4]. The proportion of canonical HOR copies to variants is 279:62 (detailed in Appendix A), thereby emphasizing the predominant classification as a 15mer. In contrast, the analysis via NTRprism, aligning with the GRM algorithm’s 2007 framework [44], interprets the same HOR configuration differently, deeming the less common 11mer as the canonical form and identifying the more frequently observed 15mer as a variant by incorporating an additional 4mer segment at the beginning of the 11mer.

Our study further revealed that several minor HORs align with the canonical Willard’s HOR. Although the functional impact of these cascade patterns has yet to be elucidated, the findings offer new insights into the complex structures of human centromeres.

It is noteworthy that our findings emphasize a significant concordance between bioinformatic analyses and traditional molecular methodologies. Specifically, the identification of a 2.5 kb band through Southern blotting, using satellite monomers as probes, aligns closely with our bioinformatic discovery of a major higher-order repeat (HOR) consisting of 15 monomers in the T2T-CHM13 assembly. This alignment not only validates the robustness of bioinformatic approaches but also demonstrates their compatibility and complementarity with conventional molecular techniques. By bridging these methodologies, our study not only advances our understanding of genomic structures but also emphasizes the importance of integrating diverse scientific approaches to uncover complex biological phenomena.

While certain HORs have been documented previously [4], this article represents a pioneering moment as it reveals their precise internal structure for the first time. Our research highlights the critical distinction between these HORs and the traditional Willard-type HORs, specifically the interspersed and cascading HORs. This discovery illuminates the complex architecture of these HORs within the centromere, shedding light on their potential role in conveying crucial genetic information, with possible implications for chromosome division and genetic stability.

## Figures and Tables

**Figure 1 ijms-25-04395-f001:**
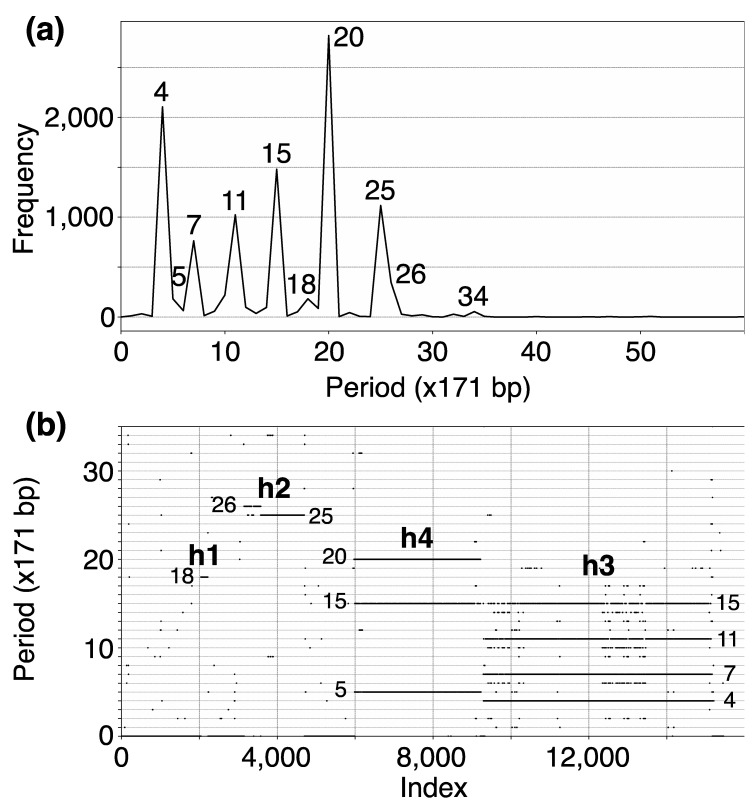
(**a**) GRM (Global Repeat Map) diagram for tandemly arranged alpha satellite monomers in the complete T2T-CHM13 assembly of human chromosome 15. Horizontal axis: GRM periods (in ~171 bp units). Vertical axis: frequency of monomer repeats period. Identified GRM peaks exhibit periods 4, 5, 7, 11, 15, 18, 20, 25, 26 and 34. The significance of these GRM peaks (HORs or associated subfragment repeats) can be inferred from the monomer distance (MD) diagram. (**b**) Monomer distance (MD) diagram for tandemly organized monomers identified in T2T-CHM13 assembly of human chromosome 15. Horizontal axis: enumeration of tandemly organized alpha satellite monomers, in sequential order as revealed via GRM analysis of the T2T assembly. Vertical axis: period (the distance between the start of a monomer and the next monomer of the same type). Four distinct regions of monomer tandems, denoted as h1, h2, h4, and h3, correlate with HORs designated as hor1, hor2, hor4, and hor3, respectively. Additionally, there are sporadic MD points that do not correspond to HORs or their subfragments.

**Figure 2 ijms-25-04395-f002:**
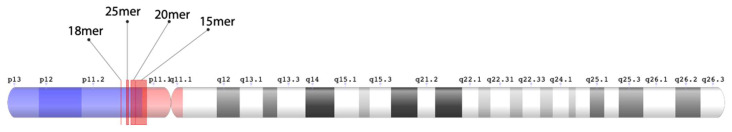
Ideogram of alpha satellite HOR arrays in human chromosome 15. All four regions of HOR arrays are located on the short (p) arm of chromosome 15. The 18mer, 25mer, and 20mer HORs are situated on the p11.2 pericentromeric subregion (light blue color). This gene-rich band is positioned on the distal portion of the p arm (p11.2q13). The array of 15mer HORs initiates within the p11.2 pericentromeric subregion and extends within the p arm heterochromatic centromeric region (red color). The positions of these HOR arrays within the T2T CHM13 genomic sequence are as follows: Willard’s type 18mer HOR array 14,699,937 bp–14,737,927 bp, Willard’s type 25mer HOR array 15,417,939 bp–15,692,443 bp, cascading 20mer HOR array 15,993,645 bp–16,555,446 bp, and cascading 15mer HOR array 16,679,039 bp–17,683,163 bp. The 20mer HOR array extends into the 15mer HOR array, while non-repetitive genetic material is present between the 18mer HOR array and the 25mer HOR array.

**Figure 3 ijms-25-04395-f003:**
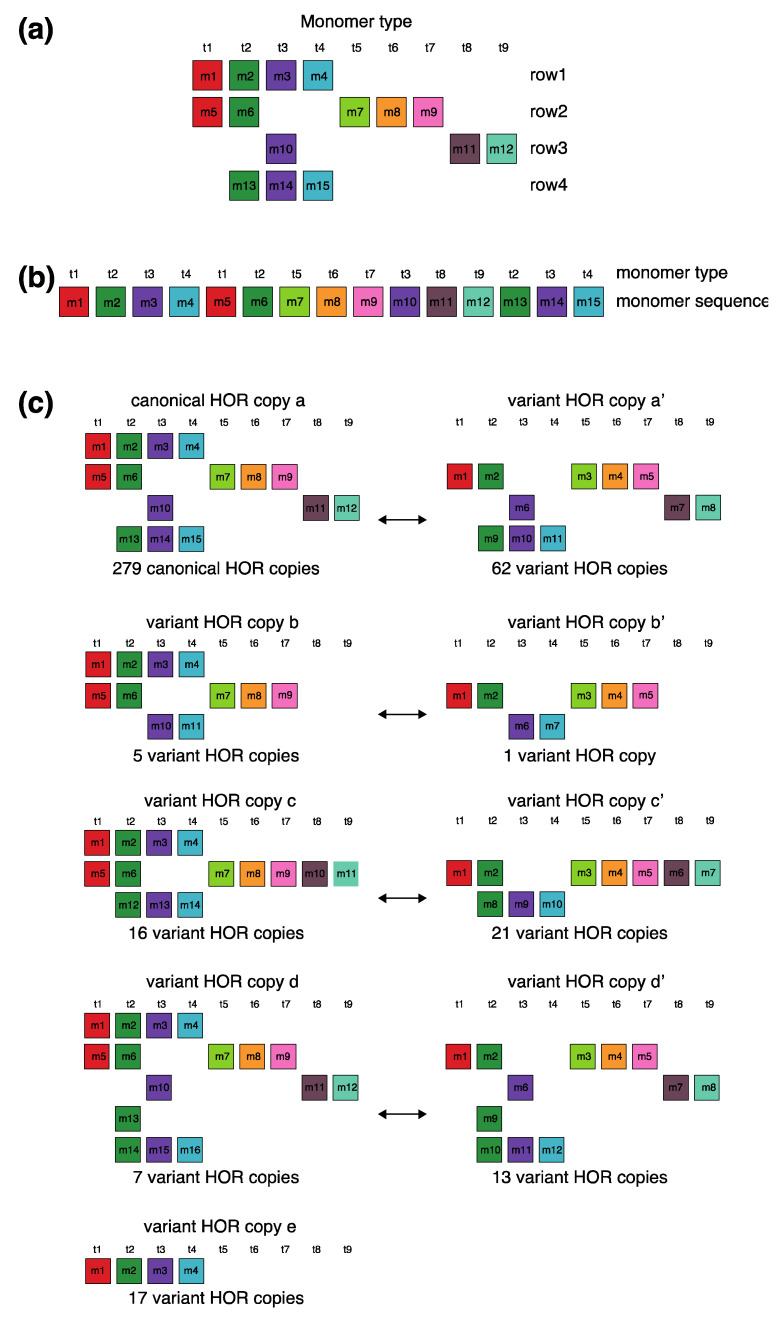
Aligned scheme for cascading 15mer HOR alignment in hor3. (**a**) Canonical HOR copy displayed in the cascading HOR presentation (n = 15, τ = 9) (15 monomers of 9 different types). Monomers within HOR copy are labeled as m1, m2, … m15, in order of their appearance within the canonical HOR copy (from left to right within each row and from top to bottom). Each monomer is depicted by a colored box, with distinct colors corresponding to different monomer types. Monomers are organized into columns based on their monomer types: monomer type t1 in the first column, monomer type t2 in the second column, and so forth. The number of columns, i.e., the number of different monomer types in the canonical HOR copy, is denoted by τ. (**b**) Attribution of 9 monomer types t1, t2, … t15 to 15 monomers m1, m2, … m15 in the one-row presentation of the canonical 15mer HOR copy. (**c**) Comparison of canonical 15mer HOR copy and some of its variants.

**Figure 4 ijms-25-04395-f004:**
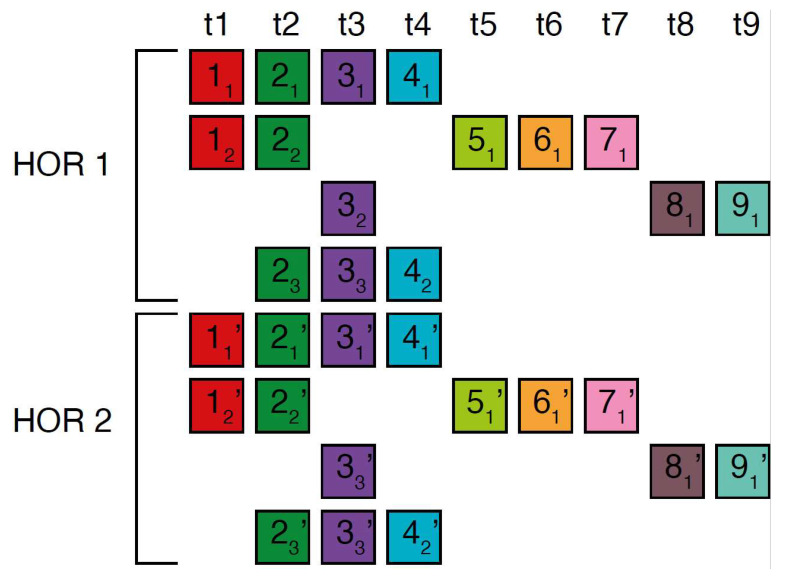
Scheme for interspersed repetitive monomeric subfragments in cascading 15mer HOR array. In the MD diagram (Figure 1b), within this same monomer enumeration interval, there exist equidistant MD line segments for periods 4, 7 and 11, which represent subfragments of the 15mer HOR. Distances between monomers of the same type are illustrated for two neighboring canonical HOR copies.In the first HOR copy, the monomers are labeld as $1_{1}$ $2_{1}$ … $4_{2},$ and in the second HOR copy as $1_{1}^{’}2_{1}^{’}$ … $4_{2}^{’}$. Here, $1_{1}$ represents the first monomer of type t1 in the first HOR copy, $2_{1}$ the first monomer of type t2 in the first HOR copy, and so forth, while $4_{2}$ denotes the second monomer of type t4 in the first HOR copy. Similarly, $1_{1}^{’}$ denotes the first monomer type t1 in the second HOR copy; $2_{1}^{’}$ denotes the first monomer of type t2 in the second HOR copy, and so on, with $4_{2}^{’}$ representing the second monomer of type t4 in the second HOR copy. Highly similar monomers are not numbered consecutively, as depicted in the figures above, to facilitate the explanation of monomer distances and MD monomeric subfragments in Figure 1b. The distance between monomers $1_{1}$ and $1_{2}$ is denoted $d(1_{1}$, $1_{2})$. This distance, $d(1_{1}$, $1_{2}),$ is equal to the sum of distances between $1_{1}$ and $2_{1},2_{1}$ and $3_{1},3_{1}$ and $4_{1},4_{1}$ and $1_{2},$ i.e., $d(1_{1}$, $1_{2})={d(1}_{1},$ $2_{1})+d(2_{1},3_{1}$) + $d(3_{1},4_{1}$) + $d(4_{1}$, $1_{2})$ ~4 monomer units. In this way, we obtain the monomer distances given in the text.

**Figure 5 ijms-25-04395-f005:**
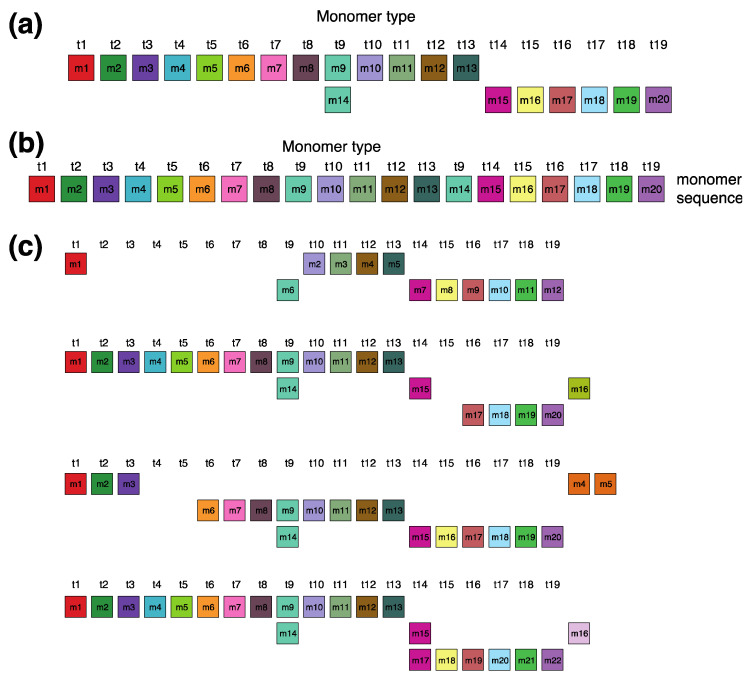
Cascading 20mer HOR alignment in hor4. (**a**) Canonical HOR copy displayed in cascading HOR presentation (n = 20, τ = 19). Monomers within the HOR copy are labeled as m1, m2, … m20, in order of appearance within canonical HOR copy (from left to right within each row and from first to second). The number of columns, i.e., the number of different monomer types in the canonical HOR copy, is denoted by τ = 19. (**b**) Attribution of 19 monomer types t1, t2, … t19 to 20 monomers m1, m2, … m20 in one-row presentation of canonical 20mer HOR copy. (**c**) Comparison of canonical 15mer HOR copy and some of its variants.

**Figure 6 ijms-25-04395-f006:**
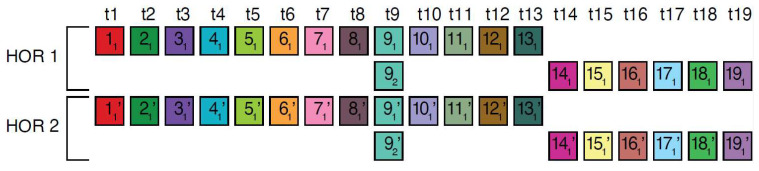
Interspersed repetitive monomeric subfragments in cascading 20mer HOR array. In the MD diagram (Figure 1b), within this same monomer enumeration interval, there exist equidistant MD line segments for periods 5 and 10, which represent subfragments of the 20mer HOR. Distances between monomers of the same type are exemplified for two adjacent canonical HOR copies. Relevant inter-monomer distances are provided for two neighboring canonical 20mer HOR copies. Relevant distances between the monomers of repeated monomers are $d\left(9_{1},9_{2}\right)=$5, ${d(9}_{1}$, $9_{1}^{,})$ = 15, giving rise to subfragment MD line segment periods 5 and 15.

**Figure 7 ijms-25-04395-f007:**
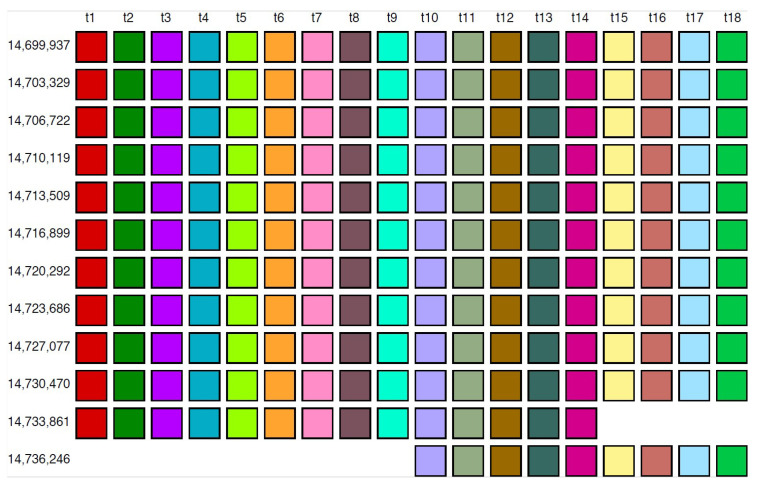
Alignment of Willard’s type 18mer HOR array.

**Figure 8 ijms-25-04395-f008:**
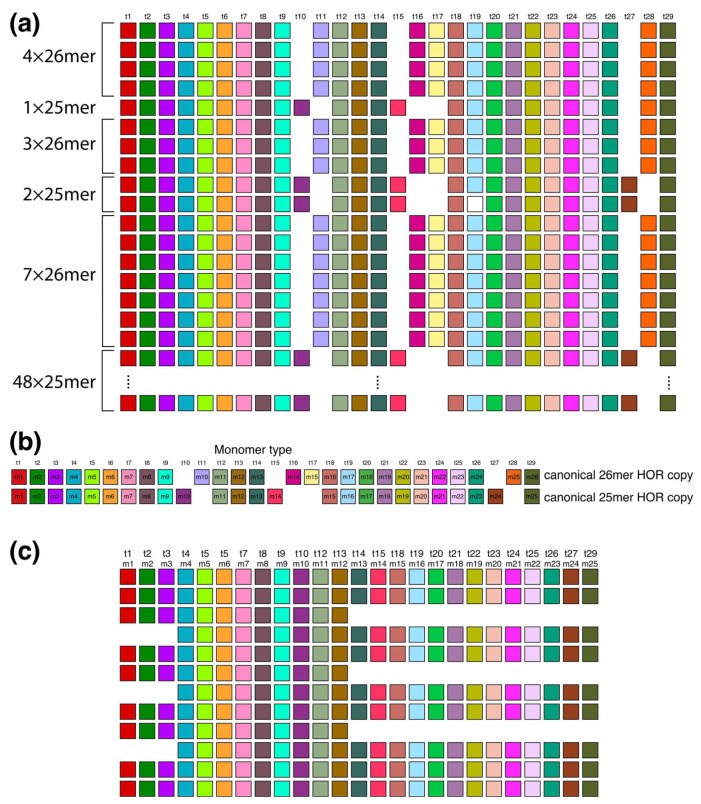
Aligned scheme of six interspersed sections of Willard’s type 25mer and 26mer HOR arrays (hor2). (**a**) From top to bottom: four 26mer HOR copies, one 25mer HOR copy, three 26mer HOR copies, two 25mer HOR copies, seven 26mer HOR copies and forty-eight 25mer HOR copies. Of the 48 25mer HOR copies at the bottom part of the figure, only the 1st and 48th 25mer HOR copies are shown. Canonical 25mer and 26mer HOR copies have 22 common t-monomers, while they differ in 7 t-monomers: in the canonical 25mer HOR copy, the missing monomers are t11, t16, t17, and t28, while in the canonical 26mer HOR copy, the missing monomers are t10, t15, and t27. Thus, the 25mer and 26mer HOR copies together comprise 29 types of HOR copies. They are all included in the t-monomer sequence at the top of the figure to demonstrate the common alignment of both 25mer and 26mer HOR copies. (**b**) Alignment of 25mer and 26mer canonical HOR copies with 29 common monomer types. (**c**) A segment of HOR copies No. 6 to 17 from the set of 48 25mer HOR copies, illustrating the origin of tertiary 34-monomer repeats (comprising three two-variant copies: No. 8–9, 11–12, and 14–15, each consisting of 12 + 22 = 34 monomers). In this graphical presentation, only the t-monomers present in canonical 25mer HOR are included (i.e., t11, t16, t17 and t28 are omitted from the t-sequence).

**Table 1 ijms-25-04395-t001:** Alpha satellite HOR arrays in T2T-chm13 assembly of human chromosome 15 determined using the GRM2023 algorithm.

HOR	Notation	*N*	*τ*	No. of HOR Copies	Type of HOR	Multimonomer Tertiary Repeat Fragments
15mer	hor3	15	9	429	Cascading	4, 7, 11
20mer	hor4	20	19	164	Cascading	5, 15
18mer	hor1	18	-	12	Willard’s type	-
25/26mer	hor2	25/26	-	51/14	Intermixed Willard’s type	34

*τ* denotes the number of different monomers in canonical cascading nmer HOR. The scheme of all HOR copies identified using the GRM algorithm are presented in Appendix A.

## Data Availability

Genomic sequences are freely available at the National Center for Biotechnology Information (NCBI) website https://www.ncbi.nlm.nih.gov. The GRM graphical user interface application (JAR file) is freely available at our project’s website http://genom.hazu.hr/tools.html, URL (accessed on 14 April 2024). It can be run on any platform that has Java Runtime Environment (JRE) installed (freely available at https://www.oracle.com/java/technologies/javase-downloads.html, URL (accessed on 14 April 2024)).

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
