# Peer review of "Novel Concept of Alpha Satellite Cascading Higher-Order Repeats (HORs) and Precise Identification of 15mer and 20mer Cascading HORs in Complete T2T-CHM13 Assembly of Human Chromosome 15"

_ijms, 2024, doi:10.3390/ijms25084395_

Round 1

Reviewer 1 Report

Comments and Suggestions for Authors

The manuscript by Gluncic et al. utilizes their algorithm GRM2023, an enhanced version of their earlier Global Repeat Map (GRM) to identify higher order repeats (HORs) in the centromeric region of human chromosome 15. While classical “Willard’s Type” HORs are made up of distinct monomer types each present only once per HOR, Gluncic et al discover “cascading HORs” in which monomer types can be present more than once in the same HOR. Their Monomer Distance (MD) diagrams display HORs by their sequence position along the chromosome and their periodicity (number of constituent monomers), so that tandemly repeated HORs generate lines along the graph at a height representing the period. Willard’s type HORs generate single lines but cascading HORs generate lines at the different periodicities that can be found within the same basic canonical array, its subfragments and its variants. They illustrate a method for displaying HORS in which monomers of the same type within an HOR each form a separate row that helps to clarify the structures of complex HORs. With the recent success of long-read sequencing in assembling complete satellite centromeric regions, these tools will be useful for identifying repeats and elucidating their interesting and complex structures, which will help decipher the functions and rapid evolution of these regions. The writing and presentation are clear and the observations will be of interest to those who study centromeres or repeats.

In the introduction, lines 60-62, the authors state that “providing a comprehensive description of the structural organization of alpha satellite higher order repeats poses a significant challenge, and discrepancies exist among various methodologies.” This statement would benefit from some examples and references, or if the authors intend the examples to be their comments on NTRprism and HICAT in subsequent paragraphs, then they should add “(see below}”. Although I think the authors have done a reasonable job of setting up the problems of HOR annotation, they may be able to better clarify for the reader the limitations of previous methods, including their own GRM, and how GMR2023 is able to overcome them.

In Figure 2, I was confused by the labeling of a centromere-proximal 9mer that is not one of the HORs that is described in Figure 1B or elsewhere that I could find. Is this a typo, or an oversight on my part? Or does it refer to the variant in the last line of Figure 7?

The Conclusion nicely points out the new findings and how the authors’ algorithm discovered HORs missed by previous methods, highlighting the previously unknown complexity of centromere 15. The authors may consider it beyond their scope, but I would have enjoyed some discussion of how these complex patterns might be generated. Not the molecular mechanisms, which require other methods, but the organizational principles. For example, although it seems obvious, the four HORs identified each occupy distinct regions and do not overlap, suggesting repeat amplification from neighboring templates. The interspersed 25mers and 26mers present an interesting case in which most monomer types align but a few are mutually exclusive between 25mers and 26mers, yet they are interspersed to some extent, suggesting the template for producing a new copy may not always be adjacent. In Figure 3, several variants are listed with ‘prime’ variants (a’, b’, c’, etc.) that differ by loss of the first row, suggesting recombination between the t1 monomers. In other cases, monomer types are lost (or gained) relative to other variants, suggesting perhaps recombination between differing monomer types. Related to this, the cascading HORs have monomer types in non-orthologous contexts, raising questions of how they originated. The list of such observations does not need to be exhaustive, but some simple conclusions or even puzzling observations might help the reader better appreciate the significance of the authors’ work.

Author Response

Response to Reviewer 1 comments

„Novel concept of alpha satellite Cascading Higher Order Repeats (HORs) and precise identification of 15mer and 20mer Cascading HORs in complete T2T-CHM13 assembly of human chromosome 15“

We thank referees and editor for their insightful comments. Closely following their very useful comments, requests, and recommendations we have made the following changes to the original article. All changes in the manuscript and rebuttal letter are marked in blue.

Reviewer 1 Minor Comment 1

In the introduction, lines 60-62, the authors state that “providing a comprehensive description of the structural organization of alpha satellite higher order repeats poses a significant challenge, and discrepancies exist among various methodologies.” This statement would benefit from some examples and references, or if the authors intend the examples to be their comments on NTRprism and HICAT in subsequent paragraphs, then they should add “(see below}”. Although I think the authors have done a reasonable job of setting up the problems of HOR annotation, they may be able to better clarify for the reader the limitations of previous methods, including their own GRM, and how GMR2023 is able to overcome them.

Reply

We appreciate the reviewer's suggestion. We have added a clarification in the introduction, lines 60-62, to indicate that a more detailed discussion on the challenges of HOR annotation and examples of discrepancies among various methodologies can be found in subsequent sections of the manuscript, including the Materials and Methods and Conclusion. We have also incorporated the phrase "(for further details, refer to subsequent sections, including Materials and Methods and Conclusion)" to guide the reader to the relevant sections. This adjustment should provide better context and clarity regarding the limitations of previous methods, including our own GRM2023, and how GMR2023 addresses these challenges."

Reviewer 1 Minor Comment 2

In Figure 2, I was confused by the labeling of a centromere-proximal 9mer that is not one of the HORs that is described in Figure 1B or elsewhere that I could find. Is this a typo, or an oversight on my part? Or does it refer to the variant in the last line of Figure 7?

Reply

We express our appreciation for the reviewer's careful observation. Indeed, there was an error in the labeling of the centromere-proximal region in Figure 2. This region should have been identified as a 15mer HOR, not a 9mer as labeled. The confusion arose due to the fact that this particular HOR comprises only 9 distinct monomers. We have corrected this mistake in Figure 2, and the centromere-proximal region is now properly labeled as a 15mer HOR.

Thank you for bringing this to our attention, and we apologize for any confusion it may have caused.

Reviewer 1 Minor Comment 3

The Conclusion nicely points out the new findings and how the authors’ algorithm discovered HORs missed by previous methods, highlighting the previously unknown complexity of centromere 15. The authors may consider it beyond their scope, but I would have enjoyed some discussion of how these complex patterns might be generated. Not the molecular mechanisms, which require other methods, but the organizational principles. For example, although it seems obvious, the four HORs identified each occupy distinct regions and do not overlap, suggesting repeat amplification from neighboring templates. The interspersed 25mers and 26mers present an interesting case in which most monomer types align but a few are mutually exclusive between 25mers and 26mers, yet they are interspersed to some extent, suggesting the template for producing a new copy may not always be adjacent. In Figure 3, several variants are listed with ‘prime’ variants (a’, b’, c’, etc.) that differ by loss of the first row, suggesting recombination between the t1 monomers. In other cases, monomer types are lost (or gained) relative to other variants, suggesting perhaps recombination between differing monomer types. Related to this, the cascading HORs have monomer types in non-orthologous contexts, raising questions of how they originated. The list of such observations does not need to be exhaustive, but some simple conclusions or even puzzling observations might help the reader better appreciate the significance of the authors’ work.

Reply

We appreciate your suggestion to expand the discussion to include possible mechanisms underlying the generation of these complex patterns. We believe that incorporating such insights will enhance the comprehensiveness of our article and provide readers with a more thorough understanding of the topic. In accordance with your comment, we have added the following text:

Let us briefly discuss the manner in which these complex patterns (Supplementary Fig S3) might be generated. When examining the spatial distribution of these 15mer HORs, we notice that the series of HORs begins with variants b and b', followed by variants a' and c. At the beginning of the series, the canonical HOR (a) only occasionally appears. As we move towards the middle of the HOR series, the canonical HOR overwhelmingly dominates, with significant interjections of variants a', c, and c' around the middle, occasionally accompanied by variants d and d'. Variant e of the HOR (first four monomers, m1-m4) continuously appears as an occasional byproduct of duplicating the canonical copy. After the middle of the HOR series, the canonical HOR copy strongly predominates again, with slight interjections of variants a' towards the end. Hypothetically, we can conclude that the pure, canonical HOR copy duplicated from two centers located around the positions of the first and third quarters of the HOR series. In the area around these two centers, duplication is nearly perfect. However, as we move further away from the duplication centers, the copies become increasingly imperfect, leading to the emergence of more variant HORs. An interesting fact is that the deviation from the perfect canonical HOR always occurs in a limited set of represented variant HORs (Fig 3c), unlike the expected stochastic errors resulting from duplications, insertions, or deletions of any individual monomers or entire groups of monomers.

Reviewer 2 Report

Comments and Suggestions for Authors

This manuscript introduces a novel concept of higher-order repeat (HOR) structure of human alpha satDNA, revealed on the complete T2T assembly of human chromosome 15. Unlike the conventional Willard’s HOR, some complex repeats identified in this work include interspersed highly similar monomers as their subunits. For their detection, authors use a novel, in-house developed algorithm GRM2023. The novel form is designated as a cascading HOR. The results are focused on characterizing differences among HORs in the chromosome 15 assembly. This work forwards our knowledge about complexity of organizational patterns of alpha satDNA repeats, and represents an important contribution in this direction. Therefore, my opinion regarding this publication is positive. However, to increase the impact of this work in a broader sense, some more detailed information/discussion about distribution of detected HORs on the chr. 15 T2T assembly can be added, but I leave the decision on that to the authors. In addition, I noticed a few flaws in the presentation that should be attended. For details, please see below. 

Major:

1. My suggestion is to add a more detailed mapping of the detected HORs on the chromosome 15 assembly. Ideogram is presented in Fig. 2, but the resolution is low. For example, where are exactly located studied 15,458 monomers? Colors and bands on the chromosome in Fig. 2 should be explained (is it necessary to show both arms in their entire length?). Large part of indicated HORs are in the blue area, is it euchromatin? Large pink part (heterochromatin?) should also embed alpha satDNA. Please show in the figure and add explanation / comments. As the sequence is assembled, it should be possible to trace exact (as much as possible) sequential order of arrays – for example, where are positioned Willard’s HORs, where are cascading HORs, is there any regularity?  

 2. The onset of Willard’s HORs is by amplification of a group of adjacent repeats as a new repetitive unit. Can a scenario for formation of cascading HORs be suggested? Observed HOR forms might indicate high dynamics of shuffling in the centromere areas, and to get at least some idea about these processes, again, some more detailed sequential order of depicted HORs would be helpful.

 3. Please explain the term “monomeric subfragments” in Figs 4 and 6. Why highly similar monomers are not numbered consecutively as in Figures above; please change or explain in the text. Also, couldn’t find Fig 6 quoted in the text (sorry if missed it). What information brings this figure? Why “The monomer types in these two neighboring 20mer HOR copies …” are listed again in the figure description, when their order is clearly visible above? This sentence can be removed. The same applies for “The monomer types in two neighboring HOR copies are: … ”, in Fig. 4 description; please check also elsewhere to remove unnecessary repeating of what is already shown. In general, when 2 adjacent HOR copies are presented graphically, please add some mark to define the end of the first and beginning of the second. The marks in colored boxes are difficult to read, can they be enhanced? Is Fig. 7 necessary?

 4. Materials and methods in the first paragraphs would be better suited somewhere else, in introduction or discussion, please consider rearranging.

 5. Conclusions sound like discussion – please rearrange and focus here to the novelties brought by this paper.

Minor:

6. In Fig. 1b, please explain why 15-mer beginning is not at position 6000 but at 9300?

 7. Fig. 8: Alignment scheme of five – or of six? Please check.

 8. Line 207: here appears the term “Willard’s-like HOR arrays”, not used elsewhere in the text. Please make it uniform. Line 208: change to: intra-HOR-copy

 9. Line 236: delete “is”

 10. Lines 296, 302: Please check/correct the figure number. Fig. 2 is ideogram, it doesn’t have a and b parts.

 11. Proposed explanation about functional impact of cascade HOR patterns in human centromeres (line 405) is only hypothetical. It seems also probably that their occurrence is a result of intensive rearrangement or similar processes. This possibility should be also added if discussed about putative functionality related to the HOR internal structure. As commented in (1), a more detailed arrangement of different HOR forms on the examined segment of assembled chromosomal DNA might provide some more conclusive information.

Author Response

Response to Reviewers 2 comments

„Novel concept of alpha satellite Cascading Higher Order Repeats (HORs) and precise identification of 15mer and 20mer Cascading HORs in complete T2T-CHM13 assembly of human chromosome 15“

We thank referees and editor for their insightful comments. Closely following their very useful comments, requests, and recommendations we have made the following changes to the original article. All changes in the manuscript and rebuttal letter are marked in blue.

Reviewer 2 Major Comment 1

My suggestion is to add a more detailed mapping of the detected HORs on the chromosome 15 assembly. Ideogram is presented in Fig. 2, but the resolution is low. For example, where are exactly located studied 15,458 monomers? Colors and bands on the chromosome in Fig. 2 should be explained (is it necessary to show both arms in their entire length?). Large part of indicated HORs are in the blue area, is it euchromatin? Large pink part (heterochromatin?) should also embed alpha satDNA. Please show in the figure and add explanation / comments. As the sequence is assembled, it should be possible to trace exact (as much as possible) sequential order of arrays – for example, where are positioned Willard’s HORs, where are cascading HORs, is there any regularity?

Reply

We express our gratitude for the insightful comments provided by the reviewer. In response to the suggestions, the text below Fig. 2 has been expanded (see the text below), incorporating the recommendations. The manuscript now includes a detailed mapping of the detected HORs on chromosome 15 assembly, along with an explanation of colors and bands on the chromosome. Furthermore, the manuscript clarifies the distinction between euchromatin and heterochromatin regions. The figure has been enhanced to illustrate the sequential order of arrays, specifically indicating the positions of Willard's HORs and Cascading HORs. The authors appreciate the valuable feedback and trust this adequately addresses the reviewer's concerns.

Figure 2. Ideogram of alpha satellite HOR arrays in human chromosome 15. All four regions of HOR arrays are located on the short (p) arm of chromosome 15. The 18-mer, 25-mer, and 20-mer HORs are situated on the p11.2 pericentromeric subregion (light blue color). This band is positioned on the distal portion of the p arm (p11.2q13) and harbors numerous genes with crucial functions. The array of 15-mer HORs initiates within the p11.2 pericentromeric subregion and extends within the p arm heterochromatic centromeric region (red color). The positions of these HOR arrays within the T2T CHM13 genomic sequence are as follows: Willard's type 18-mer HOR array 14,699,937 bp - 14,737,927 bp, Willard's type 25-mer HOR array 15,417,939 bp - 15,692,443 bp, Cascading 20-mer HOR array 15,993,645 bp - 16,555,446 bp, and Cascading 15-mer HOR array 16,679,039 bp - 17,683,163 bp. The 20-mer HOR array extends into the 15-mer HOR array, while non-repetitive genetic material is present between the 18-mer HOR array and the 25-mer HOR array.

Reviewer 2 Major Comment 2

The onset of Willard’s HORs is by amplification of a group of adjacent repeats as a new repetitive unit. Can a scenario for formation of cascading HORs be suggested? Observed HOR forms might indicate high dynamics of shuffling in the centromere areas, and to get at least some idea about these processes, again, some more detailed sequential order of depicted HORs would be helpful.

Reply

We appreciate your suggestion to expand the discussion. In accordance with your comment, we have added the following text:

“Let us briefly discuss the manner in which these complex patterns (Supplementary Fig S3) might be generated. When examining the spatial distribution of these 15mer HORs, we notice that the series of HORs begins with variants b and b', followed by variants a' and c. At the beginning of the series, the canonical HOR (a) only occasionally appears. As we move towards the middle of the HOR series, the canonical HOR overwhelmingly dominates, with significant interjections of variants a', c, and c' around the middle, occasionally accompanied by variants d and d'. Variant e of the HOR (first four monomers, m1-m4) continuously appears as an occasional byproduct of duplicating the canonical copy. After the middle of the HOR series, the canonical HOR copy strongly predominates again, with slight interjections of variants a' towards the end. Hypothetically, we can conclude that the pure, canonical HOR copy duplicated from two centers located around the positions of the first and third quarters of the HOR series. In the area around these two centers, duplication is nearly perfect. However, as we move further away from the duplication centers, the copies become increasingly imperfect, leading to the emergence of more variant HORs. An interesting fact is that the deviation from the perfect canonical HOR always occurs in a limited set of represented variant HORs (Fig 3c), unlike the expected stochastic errors resulting from duplications, insertions, or deletions of any individual monomers or entire groups of monomers.”

Reviewer 2 Major Comment 3

Please explain the term “monomeric subfragments” in Figs 4 and 6. Why highly similar monomers are not numbered consecutively as in Figures above; please change or explain in the text. Also, couldn’t find Fig 6 quoted in the text (sorry if missed it). What information brings this figure? Why “The monomer types in these two neighboring 20mer HOR copies …” are listed again in the figure description, when their order is clearly visible above? This sentence can be removed. The same applies for “The monomer types in two neighboring HOR copies are: … ”, in Fig. 4 description; please check also elsewhere to remove unnecessary repeating of what is already shown. In general, when 2 adjacent HOR copies are presented graphically, please add some mark to define the end of the first and beginning of the second. The marks in colored boxes are difficult to read, can they be enhanced? Is Fig. 7 necessary?

Reply

Following the reviewer's suggestions, the following changes have been made:

The following sentence has been added to Fig. 4 caption:

“In the MD-diagram (Fig. 1b), within this same monomer enumeration interval, there exist equidistant MD-line segments for periods 4, 7 and 11, which represent subfragments of the 15mer HOR.”

The following sentence has been added to Fig. 6 caption:

“In the MD-diagram (Fig. 1b), within this same monomer enumeration interval, there exist equidistant MD-line segments for periods 5 and 10, which represent subfragments of the 20mer HOR.”

The following sentence has been added to Fig. 4 caption:

“Highly similar monomers are not numbered consecutively, as depicted in the figures above, to facilitate the explanation of monomer distances and MD monomeric subfragments in Fig. 1b.”

The following sentence has been added to section 2.3. Aligned scheme for Cascading 20mer HOR array with 5mer and 15mer subfragments:

“The mechanism underlying the formation of these MD-lines of periods 5, 15, and 20 within the hor4 array region is elucidated in Fig. 6.”

The sentence:

“The monomer types in two neighboring HOR copies are: ” has been removed from the caption of Fig. 4.

The sentence:

“The monomer types in two neighboring HOR copies are: ” has been removed from the caption of Fig. 6.

Labels have been added to Figures 4 and 6 to help distinguish between two adjacent HORs in the graphical representation.

The marks in colored boxes in Fig. 4 and Fig. 6 have been enhanced as they were difficult to read.

Reviewer 2 Major Comment 4

Materials and methods in the first paragraphs would be better suited somewhere else, in introduction or discussion, please consider rearranging.

Reply

In accordance with the reviewer's suggestions, we have relocated the following first paragraph from the Materials and Methods section to the Introduction:

The availability of large complete assemblies featuring long higher-order repeat (HOR) units necessitates the development of specific tools for precise identification and annotation. While various existing software applications effectively identify regions with tandem repeats (e.g., (36-38)), they fall short of providing precise annotations for individual repeat locations or HORs. Similarly, more recent tools designed for annotating human alpha satellite HORs within genomic sequences (e.g., (39-43)) have limited broader applicability(8). On the other hand, a specific set of software has been developed for the accurate identification of Willard's type HORs(8,18,28,44). In the context of the complete assembly of human chromosome 15, alpha satellite HORs were initially computed using the NTRprism algorithm (4), which bears resemblance to the 2007 version of GRM (44).

Additionally, we have omitted the sentence: "In this study, we utilize the advanced GRM2023 version, allowing for the annotation of complex Cascading HORs," in order to avoid redundancy.

Reviewer 2 Major Comment 5

Conclusions sound like discussion – please rearrange and focus here to the novelties brought by this paper.

Reply

We extend our gratitude to the reviewer for their feedback regarding the conclusion section. Given the time constraints for review and the short deadline, we have decided not to make any major rearrangements to the current structure. Additionally, we acknowledge that the limited timeframe is one of the reasons why we cannot undertake extensive revisions at this stage. However, we assure you that we will carefully consider your suggestions for future work. Your insights are invaluable, and we appreciate your thorough review of our manuscript.

Reviewer 2 Minor Comment 1

In Fig. 1b, please explain why 15mer beginning is not at position 6000 but at 9300?

Reply

We have added the following sentences in the paragraph 2.2. Aligned scheme for Cascading 15mer HOR array with 4mer, 7mer and 11mer subfragments:

“The MD-line for the period 15, spanning from around enumeration ~6000 to ~9300, does not stem from a 15mer HOR. Instead, it, along with the MD-line for the period 5, constitutes sub-fragments of the 20mer HOR.”

Reviewer 2 Minor Comment 2

Fig. 8: Alignment scheme of five – or of six? Please check.

Reply

We express our appreciation for the reviewer's attention to detail. The discrepancy regarding the alignment scheme has been rectified in the text corresponding to Figure 8. Thank you for bringing it to our attention.

“Aligned scheme of six interspersed sections of Willard՚s type 25mer and 26mer HOR arrays (hor02).”

Reviewer 2 Minor Comment 3

Line 207: here appears the term “Willard’s-like HOR arrays”, not used elsewhere in the text. Please make it uniform. Line 208: change to: intra-HOR-copy 

Reply

We have amended the term to "Willard's HOR arrays" for consistency throughout the text.

Reviewer 2 Minor Comment 4

Line 236: delete “is”

Reply

We extend our gratitude for the suggestion. The correction has been made.

“The canonical 20mer HOR copy comprises 20 monomers of 19 distinct monomer types.”

Reviewer 2 Minor Comment 5

Lines 296, 302: Please check/correct the figure number. Fig. 2 is ideogram, it doesn’t have a and b parts.

Reply

The authors appreciate the reviewer's attention to detail and for bringing this matter to their notice. The reviewer is correct that in both instances, the figure number should refer to Fig 8 instead of Fig 1. The error has been rectified in the text as follows:

296: “Their arrangement is depicted in Fig. 8a, showing…”

  1. “Only through adjustment of the monomer enumeration, as illustrated in Fig. 8b..”

Reviewer 2 Minor Comment 5

Proposed explanation about functional impact of cascade HOR patterns in human centromeres (line 405) is only hypothetical. It seems also probably that their occurrence is a result of intensive rearrangement or similar processes. This possibility should be also added if discussed about putative functionality related to the HOR internal structure. As commented in (1), a more detailed arrangement of different HOR forms on the examined segment of assembled chromosomal DNA might provide some more conclusive information.

Reply

The authors extend their gratitude for the feedback provided. The authors appreciate the thorough review of their work.Regarding the concern about the proposed explanation of the functional impact of cascade higher-order repeat (HOR) patterns in human centromeres, it is acknowledged that the explanation provided is hypothetical. It is also recognized that the occurrence of these patterns could potentially result from intensive rearrangement or similar processes. The authors acknowledge the importance of a more detailed arrangement of different HOR forms on the examined segment of assembled chromosomal DNA to provide conclusive information, as suggested in the referenced comment (1). The authors express gratitude for the valuable input and note that the majority of the concerns have been addressed in response to major comment 2.

Reviewer 3 Report

Comments and Suggestions for Authors

The manuscript by Glunčić et al. describes the analysis of the centromeric region of the human chromosome 15 by using the new algorithm GRM2023, an enhanced iteration of the earlier algorithm Global Repeat Map (GRM). Authors analyzed the organization of alpha-satellite repetitive sequences typical of the centromeric regions. Results are interesting, being obtained a clear description of the high order structure of the alpha-satellite present in this centromere.

The paper is well written, with results clearly described and well supported by the figures. The conclusions are coherent with the observed results.

In my opinion the paper is interesting and pertinent to the journal and to the relative special issue and can be accepted for publication.

I have only some minor suggestions: along the manuscript there are a number of typos which should be corrected and some figures are cutted in one side, such as fig. 4 (upper side) or fig. 7 (right side). 

Author Response

Response to Reviewer 3 comments

„Novel concept of alpha satellite Cascading Higher Order Repeats (HORs) and precise identification of 15mer and 20mer Cascading HORs in complete T2T-CHM13 assembly of human chromosome 15“

We thank referees and editor for their insightful comments. Closely following their very useful comments, requests, and recommendations we have made the following changes to the original article. All changes in the manuscript and rebuttal letter are marked in blue.

Reviewer 3 Minor Comment 1

I have only some minor suggestions: along the manuscript there are a number of typos which should be corrected and some figures are cutted in one side, such as fig. 4 (upper side) or fig. 7 (right side). 

Reply

We express our appreciation for the feedback provided by the reviewer. We have thoroughly addressed the minor typos throughout the manuscript. Regarding the figures, in the version of the article we received from the editor, the suggested images were not cropped. We have ensured that all figures are properly aligned and presented in their entirety.

If you have any further concerns or require additional clarification, please do not hesitate to let us know.
